# Replacing Mancozeb with Alternative Fungicides for the Control of Late Blight in Potato

**DOI:** 10.3390/jof9111046

**Published:** 2023-10-25

**Authors:** Yariv Ben Naim, Yigal Cohen

**Affiliations:** Faculty of Life Sciences, Bar Ilan University, Ramat Gan 529000, Israel; yar2710@gmail.com

**Keywords:** CAA fungicides, cyazofamid, disease control, EU commission, mancozeb non-renewal approval, mandipropamid, oomycetes, orondis, oxathiapiprolin, oxysterol binding proteins, tomato, Zorvec

## Abstract

Mancozeb (MZ) is a broadly used fungicide for the control of plant diseases, including late blight in potatoes caused by the oomycete *Phytophthora infestans* (Mont.) De Bary. MZ has been banned for agricultural use by the European Union as of January 2022 due to its hazards to humans and the environment. In a search for replacement fungicides, twenty-seven registered anti-oomycete fungicidal preparations were evaluated for their ability to mitigate the threat of this disease. Fourteen fungicides provided good control (≥75%) of late blight in potted potato and tomato plants in growth chambers. However, in Tunnel Experiment 1, only three fungicides provided effective control of *P. infestans* in potatoes: Cyazofamid (Ranman, a QiI inhibitor), Mandipropamid (Revus, a CAA inhibitor), and Oxathiapiprolin + Benthiavalicarb (Zorvek Endavia, an OSBP inhibitor + CAA inhibitor). In Tunnel Experiment 2, these three fungicides were applied at the recommended doses at 7-, 9-, and 21-day intervals, respectively, totaling 6, 4, and 2 sprays during the season. At 39 days post-inoculation (dpi), control efficacy increased in the following order: Zorvec Endavia > Ranman > Revus > Mancozeb. Two sprays of Zorvec Endavia were significantly more effective in controlling the blight than six sprays of Ranman or four sprays of Revus. We, therefore, recommend using these three fungicides as replacements for mancozeb for the control of late blight in potatoes. A spray program that alternates between these three fungicides may be effective in controlling the disease and also in avoiding the build-up of resistance in *P. infestans* to mandipropamid and oxathiapiprolin.

## 1. Introduction

Mancozeb (MZ) is an ethylenebisdithiocarbamate (EBDC) non-systemic agricultural fungicide with a multi-site protective action on contact.

MZ controls numerous fungal diseases in a wide range of field crops, fruits, nuts, vegetables, and ornamentals, including late blight in potatoes and tomatoes [1]. Its mode of action (MOA) involves interference with enzymes having sulfhydryl groups, disrupting biochemical processes within the fungal cell cytoplasm and mitochondria [2]. MZ is one of the most commonly used fungicides in the world, having been on the market since 1962. MZ market size in 2022 was 795.4 million US dollars [3], while in Minnesota alone, 1.1 million lb of MZ were sold in 2017 [4]. As a Group M (multi-site) fungicide, MZ has a low risk of resistance development due to its multi-site activity on fungal pathogens. It is often tank-mixed with single-site fungicides to help with resistance management [5].

One of the degradation products of MZ in the soil is ethylenethiourea (ETU). ETU, a group B2 carcinogen, was found to be a drinking water pollutant resulting from the use of EBDCs [6]. MZ can cause human health problems, including hepatic, renal, genotoxic, and hematological disorders [7,8,9,10]. EFSA concluded that MZ is likely to meet the criteria for endocrine disruption in non-target organisms. It poses a high risk to birds, mammals, non-target arthropods, and soil macroorganisms [11].

These environmental and human health risks led the European Union to issue a non-renewal approval for MZ (European Union Commission Implementing Regulation (EU) 2020/2087) [12]. The Commission forced the EU member states to withdraw all authorizations for plant protection products containing MZ until July 2021. The grace period for farmers to use up any already-bought stocks ended in January 2022 [13].

This non-renewal approval for MZ has a dramatic effect on farmers, especially those who export their produce to the EU. Israeli farmers who export fresh potato tubers to the EU market every spring now look for effective alternative MZ-free fungicides for the control of late blight in potatoes caused by the oomycete *P. infestans*.

Dithiocarbamate-type compounds became key tools for the management of fungal plant diseases. Tetramethylthiuram disulfide, more commonly known as thiram, was demonstrated to be an effective seed dressing by Muskett and Colhoun [14], and Harrington [15] demonstrated the utility of thiram for the control of turf diseases.

Thiram was not effective enough when applied as a foliar spray, and the next generation of more active molecules, based on metal salts of dithio-carbamic acid, appeared soon. Ferric dimethyl dithiocarbamate (ferbam) was first reported by Anderson [16] and by Kincaid [17]. It provided good control of orchard diseases and gained wide acceptance as a spray for ornamentals due to its lower phytotoxicity compared to copper or sulfur sprays. Following ferbam was the closely related ziram (zinc dimethyldithiocarbamate), which was more useful in vegetable crops [18,19]. Nabam can be considered the first true ethylenebisdithiocarbamate (EBDC). It was unstable, with variable performance [20]. New EBDCs continued to appear, and DuPont was granted a patent for manganese ethylene bisdithiocarbamate (maneb) in 1950 [21]. Maneb was more active than Nabam or Zineb and raised the bar for performance yet further. In 1962, Rohm and Haas registered mancozeb (MZ), the zinc ion complex of maneb, which became the most important and commercially significant of all EBDCs. Two alkylene bisdithiocarbamate fungicides were also developed around the same time: propineb and metiram [22]. By the mid-1960s, the EBDC fungicides were considered the most important and versatile group of organic fungicides yet discovered [23]. These compounds worked effectively to control numerous fungal plant pathogens such as Oomycetes, Ascomycetes, Deuteromycetes, Basidiomycetes, Bacteria, and other diseases of more than 70 crops [5].

The major crops and major known diseases controlled by mancozeb are grapevine downy mildew caused by *Plasmopara viticola*, black rot caused by *Guignardia bidwellii*, rotbrenner caused by *Pseudopezicula tracheiphila*, phomopsis caused by *Phomopsis viticola*, citrus anthracnose caused by *Colletotrichum* spp., black spot caused by *Guignardia citricarpa*, melanose caused by *Diaporthe citri*, brown rot caused by *Phytophthora* spp., banana black sigatoka caused by *Mycosphaerella fijiensis*, cucurbits downy mildew caused by *Pseudoperonospora cubensis*, anthracnose caused by *Colletotrichum orbiculare*, alternaria caused by *Alternaria alternata*, gummy stem blight caused by *Didymella bryoniae*, bacterial diseases caused by *Xanthomonas* campestris pv., cucurbitae and *Pseudomonas syringae,* corn rust caused by *Puccinia sorghi*, corn leaf blight caused by *Helminthosporium maydis*, wheat leaf spot caused by *Mycosphaerella graminicola* and peanut Cercospora leaf spot caused by *Cercospora arachidicola* [5].

MZ effectively controlled diseases of potato and tomato, including late blight caused by *Phytophthora infestans* [24,25,26,27,28,29,30,31], early blight caused by *Alternaria solani* [32,33], leaf spot *Septoria lycopersici* [34,35], leaf mold caused by *Cladosporium fulvum* [36], anthracnose caused by *Colletotrichum coccodes* [37,38], bacterial speck caused by *Pseudomonas syringae* [39] and bacterial spot caused by *Xanthomonas* spp. [40,41,42].

In the last decades, additional chemical compounds have been introduced to the market for the control of plant diseases, including copper-based fungicides, bio-control agents, and chemical inducers [43,44,45]. *Phytophthora infestans* was the major factor in the infamous 1846 Irish famine and is still a major cause of heavy yield losses to potato production worldwide [1,46]. Yield losses caused by late blight and the cost of its control measures have been estimated to exceed six billion euros annually [47]. To prevent late blight, growers apply fungicides on a weekly basis with mixtures of protective and systemic fungicides [5,31].

The purpose of the present study was to evaluate all 27 anti-oomycete fungicides (and mixtures) registered in Israel for their ability to control late blight in potatoes and tomatoes. Our study was conducted in three steps: evaluating 27 fungicides in potted potato and tomato plants in growth chambers; selecting the best-performing fungicides for Tunnel Experiment 1; and evaluating the efficacy of the best fungicides in Tunnel Experiment 2. This study was focused on multi-site and single-site fungicides that may serve as adequate alternatives to MZ for the control of late blight in potatoes.

## 2. Materials and Methods

### 2.1. Growth Chambers Experiments

Tomato *cv* Roter Gnom and potato *cv* Sifra were used. Tomato plants were grown from seeds in 100 mL pots, while potato plants were grown from tubers in 250 mL pots. At the ten-leaf stage, plants were sprayed onto their upper leaf surface with a fungicide suspension containing 1, 10, 100, or 1000 ppm (product) and 3 h later inoculated with sporangial suspension (5 × 10^3^ sporangia per mL) of isolate 164 of *P. infestans* as described before [48]. At 7 dpi, the disease control efficacy of a fungicide was determined according to the leaf area occupied by late blight lesions relative to control plants, as described elsewhere [49].

### 2.2. Tunnel Experiment 1

#### 2.2.1. Plants

Potato cv Sifra were sown on 23 October 2022 in one hundred twenty-four polystyrene containers (120 L, 120 × 50 × 20 cm) filled with compost: peat (1:10, *v*/*v*), five tubers per container. The containers were arranged in four rows, with 31 containers per row. Containers were placed in a 50 × 7 × 4 m net house (covered with a white plastic net of fifty mesh) located at the Experimental Farm of Bar Ilan University, Ramat Gan, Israel (3204.1519, N, 03450.5853, E). On 13 November 2022, 21 days after sowing, when plants were ~20 cm tall and developed 10–12 compound leaves, they were sprayed with fungicides with the aid of an electric hand sprayer, ~10 mL per plant. A list of the 14 fungicides, their dose, and their concentration are shown in Table 1. Approximately 2 days after spray, plants were inoculated with a sporangial suspension of *P. infestans* (5 × 10^3^ sporangia per mL).

#### 2.2.2. Fungicides

The fourteen fungicides used in Tunnel Experiment 1 are listed in Table 1. They were selected from a total of twenty-seven fungicides due to their adequate control efficacy against late blight in potted tomato and potato plants in the growth chamber (not shown). Of the fourteen fungicides, nine were composed of two active ingredients with different modes of action. Plants were sprayed with the aid of an electric hand sprayer, with ~10 mL of fungicidal suspension per plant. Eight random replicates of one container, each with five plants, were used for each fungicide. Control plots were left untreated with fungicides.

#### 2.2.3. Pathogen

A mixture of isolates of *P. infestans* was used for inoculation. Isolates were collected from infected potato fields in Western Negev Israel during 2020, 2021, and 2022 and propagated since then on detached tomato leaves in growth chambers at 18 °C. Isolates belonged to genotypes 23A1 (resistant to mefenoxam) and 36A2 (sensitive to mefenoxam), having a simple or composed virulence structure [49]. Plants were inoculated at 5 pm with two liters of sporangial suspension containing 5 × 10^3^ sporangia per mL. To ensure a successful infection, the inoculated plants were immediately covered with transparent plastic sheets until 8 am the following morning. The temperature at night ranged between 15 and 18 °C.

### 2.3. Tunnel Experiment 2

Tunnel Experiment 2 was designed in a similar manner as Tunnel Experiment 1, with some changes. The potato cultivar used was Rozana; sowing took place on 13 November 2022; four randomly arranged replicate plots with six containers each, with five plants each, were used per fungicide. Fungicides were first applied on 8 December 2022, 25 days after sowing, when plants had 10–12 compound leaves. Inoculation with mixed isolates (as before) was carried out on 8 December 2022 at 5 pm.

Three fungicides were used in Tunnel Experiment 2, based on the results that were obtained from Tunnel Experiment 1. MZ served as a positive control. They were applied at 7-, 9-, or 21-day intervals, as shown in Table 2. Control plots were left untreated with fungicides.

### 2.4. Disease Assessment and Fungicide Efficiency

Six disease records (at 6, 8, 10, 12, 14, and 21 dpi) were taken during the epidemic period of Tunnel Experiment 1, and seven disease records (at 6, 12, 18, 24, 27, 33, and 39 dpi) during the epidemic period of Tunnel Experiment 2. Disease records were taken as a visual estimation of the proportion of infected leaf area in each plot (5 plants).

The Area Under Disease Progress Curve (AUDPC) was calculated using the % infected leaf area.
$$\mathrm{AUDPC}=\overset{n-1}{\underset{i=1}{\sum }}\left(\frac{y_{i}+y_{i+1}\;}{2}\right)(t_{i+1}-t_{i})$$
where *y_i_* is % infected leaf area at time *i*, *n* is the number of records taken, and *t* is the number of days between *t_i_* and *t_i+_*_1_.

### 2.5. Potato Yield

At day 65 post-sowing, the tubers of each treatment were bulk collected from the soil and weighted.

### 2.6. Data Analysis

Growth chamber experiments were repeated twice, with three replicate plants per fungicide dose. Tukey’s HSD (honestly significant difference) test was performed to detect significant differences at α = 0.05 between the mean efficacy of fungicides. Two Tunnel Experiments were carried out in a randomized complete block design. In Tunnel Experiment 1, eight replicates of one container (plot) with five plants each were used. In Tunnel Experiment 2, four replicates of six containers (plots) with five plants in each plot were used. Tukey’s HSD test was employed to determine if differences in mean disease records between treatments are significant at α = 0.05.

## 3. Results

### 3.1. Growth Chamber Experiments

Thirteen fungicides out of twenty-seven showed poor control efficacy for the disease at 1000 ppm. Fourteen fungicides were effective, showing partial control at 10 ppm and full control of the disease at 100 ppm. Figure 1 presents the efficacy of various doses of one fungicide (ZE) in controlling the blight in potted potato and tomato plants.

### 3.2. Tunnel Experiment 1

The experimental design of the tunnel experiment is shown in Figure 2A,B. In Tunnel Experiment 1, late blight symptoms appeared as early as 6 dpi (Figure 2C). They were observed in control plots (66.6 ± 10.4% infected leaf area), in plots treated with product 12 (polyram, 32.3 ± 19.0%), and in plots treated with product 1 (valifenalate + copper, 3.1 ± 2.5%), but not in the other treatments (Figure 2D). At 14 dpi, the best-performing products were 8 (Revus), 10 (Ranman), and 13 (Zorvek Endavia) (Figure 3). The percentage of infected leaf area in plots treated with these products was 18.5 ± 15.3, 18.6 ± 14.3, and 18.2 ± 16.3, respectively, whereas control pots exhibited 97.7 ± 8.9% infected leaf area. The percentage of infected leaf areas in other plots ranged between 27.8 and 70.8% (Figure 3). The statistical analysis confirms that products 8 (REV), 10 (RAN), and 13 (ZE) were better protective compared to the other products. (Figure 3).

Figure 4 represents the efficacy of the fourteen fungicidal products on disease intensity at 21 dpi. Product 13 (ZE) was outperforming, providing 66% protection against the late blight. Products 1, 2, 3, 7, and 9 were poorly effective (8–27% protection), while products 4,5, 6, 8, 10, 11, and 14 were moderately effective (42–60% protection) (Figure 4).

The distribution of tuber yields at 55 days after sowing and AUDPC at 21 dpi are shown in Figure 5. The data show a negative association between yield and AUDPC. The smallest AUDPC values were provided by products 10 (RAN), 8 (REV), and 13 (ZE). Plots treated with these products were among the best yielders (Figure 5).

### 3.3. Tunnel Experiment 2

Late blight symptoms first appeared at 6 dpi (Figure 6). The percent infected leaf area in control plots and in plots treated with MZ, RAN, REV, and ZE was 23.9 ± 8.5, 1.9 ± 2.2, 1.8 ± 2.0, 2.5 ± 3.0, and 0.1 ± 0.1%, respectively, indicating that ZE provided significantly better control of the disease than the other fungicides at this time. The progress of the disease from 6 dpi to 39 dpi is shown in Figure 6. At 18 dpi, the disease reached a level of 96 ± 13, 36 ± 22, 27 ± 17, 29 ± 22, and 5.7 ± 7.6% infected leaf area in plots treated with none, MZ, RAN, REV, and ZE, respectively, with ZE providing significantly better control of the disease than the other fungicides at this time (Figure 6). At 39 dpi, at the end of the experiment, ZE and RAN did not differ significantly in the level of protection they provided, 68.9% and 71.3%, respectively, but were significantly different from the other two fungicides (Figure 6).

The distribution of AUDPC for the various treatments is shown in Figure 7A. Their values reached 2976, 1073, 754, 918, and 400 units in the control, MZ, RAN, REV, and ZE-treated plots, respectively (Figure 7A). Statistical differences were in the order ZE > RAN > REV > MZ > control (Figure 7A).

Tubers were harvested 65 days after sowing. The tuber weight distribution is shown in Figure 7B. While only 0.5 kg of tubers were harvested from each control plot, 5.4, 6.2, 4.8, and 6.6 kg of tubers were harvested from MZ, RAN, REV, and ZE-treated plots, respectively (Figure 7B). ZE provided a significantly higher yield than REV.

## 4. Discussion

Based on the toxicological effects of MZ on humans and the environment [6,7,8,9,10,11], the European Commission published in December 2020 Regulation 2020/2087, declaring the non-renewal approval of the active substance MZ. Farmers were allowed to use up any already-purchased stocks until January 2022 [12].

On the other hand, the EU Commission published in May 2021 an extension of the approval period for several active substances, including benthiavalicarb, captan, cymoxanil, dimethomorph, famoxadone, folpet, and propamocarb [13].

MZ is among the most commonly used fungicides in the world, and farmers consider it crucial for plant disease control and fungicide resistance management. MZ is important to many crops exported to the EU from elsewhere (potatoes and vegetables). The farmers face the challenge of finding alternative fungicides that are as effective, available, and affordable as MZ.

Because potato is a major crop exported from Israel to the EU and late blight is a major disease of potato [31,46,47], we have launched the present study in which twenty-seven anti-oomycete, registered fungicides were evaluated for their efficacy in controlling late blight in potato in growth chambers. The list of fungicides (Table 1) included single fungicides or mixed fungicides, which belong to various chemical groups [2].

Only fourteen out of twenty-seven registered fungicidal preparations showed good capacity to control the disease (≥75%) in potted potato or tomato plants in growth chambers. Those were evaluated in Tunnel Experiment 1 with a single spray application. The three best-performing products were further examined in Tunnel Experiment 2, where they were applied a few times during the season at various time intervals, with MZ serving as another control.

The results obtained in Tunnel Experiment 1 allowed us to select the best fungicides according to their highest efficacy, namely REV, RAN, or ZE.

The efficacy of REV, RAN, and ZE was compared to that of MZ in Tunnel Experiment 2. Here, the multisite fungicides RAN and MZ were applied once a week, whereas the site-specific fungicides REV and ZE were applied once every 9 and 21 days, respectively. At 12 and 18 dpi, disease severity in control plots reached 77.3% and 95.9%, respectively, while a single preventive spray of ZE allowed the disease to colonize only 0.4% and 5.7% of the foliage, respectively, reaffirming the prolonged control efficacy of ZE [48,49,50]. In both cases, REV, RAN, and MZ were significantly less effective than ZE. As the disease progressed with time, the efficacy of ZE gradually declined while that of the other fungicides was maintained.

Lu et al. [51] reported similar results with potato late blight experiments. Mancozeb provided 72.8% protection compared with the control, whereas oxathiapiprolin, dimethomorph, or fluopicolide + propamocab provided 83.9–90.3% control of the disease. Khadka et al. [52] reported 90% efficiency of dimethomorph in controlling the late blight of potatoes, while the fenamidone/mancozeb mixture and mancozeb reached 68% and 47% control, respectively.

Mitani et al. [27] reported that cyazofamid provided excellent control of late blight in the field. It was superior to that of Mancozeb Dowley and Osullivan studied the chemical control of late blight in potato fields and concluded that the relative effects of the fluazinam, mancozeb, and phenylamide-based programs on total and marketable yields were inconsistent and varied between years [53].

Najdabbasi et al. [54] reported on the synergy between phosphite and other fungicides in the control of late blight in the field. They concluded that phosphites can boost the inhibitory activity of fungicides more than when they are applied alone.

Ivanov et al. [31] summarized the basic approaches to fighting late blight in the field: fungicides; R-gene-based resistance of potato species; RNA interference approaches; and other approaches. Babli et al. [55] reviewed the efficacy of bio-control agents against the late blight of potatoes. *Trichoderma*, *Pseudomonas*, and *Chaetomium* gave good control of the disease.

Gopi et al. [45] showed that garlic, *Trichoderma harzianum*, copper oxychloride, and copper hydroxide were effective against late blight, and these were proposed as alternatives to commercial fungicides.

Similarly, with the control of downy mildew *P. cubensis* in cucumber, oxathiapiprolin, propamocarb, and cyazofamid provided the highest reduction of disease while maintaining high yields [56].

The data obtained in Tunnel Experiment 2 confirmed that two sprays of ZE are as effective as six sprays of RAN, six sprays of MZ, or four sprays of REV. The fewer ZE sprays may compensate for their higher cost. In Israel, the cost ratio of MZ:ZE is 1:6.

The risk of oxathiapiprolin resistance development in *P. infestans* and other oomycete fungal pathogens is medium to high, and strict resistance management measures are required [57,58]. Indeed, two sprays of ZE, alternated with fungicides with different MOAs, are recommended by the manufacturer to avoid building up resistance against oxathiapiprolin [58]. No resistance has yet been reported for Benthiavalicarb [2,59]. Four sprays of REV (mandipropamid) produced no resistance to REV in *P. infestans* in our previous field studies [60]. They were reaffirmed by FRAC [2], especially if combined with other MOAs in the spray program. The recent report on the resistance of *P. infestans* EU43 to mandipropamid in Denmark [61] reaffirms that this fungicide should not be used alone but in mixtures or alternations with other MOAs.

We conclude that RAN, REV, and ZE may serve as efficient alternative fungicides for the control of late blight in field-grown potatoes. A spray program that alternates between the three fungicides may be effective in controlling the disease but also in avoiding the buildup of resistance of *P. infestans* to the single-site fungicides oxathiapiprolin, mandipropamid, and benthiavalicarb.

## Figures and Tables

**Figure 1 jof-09-01046-f001:**
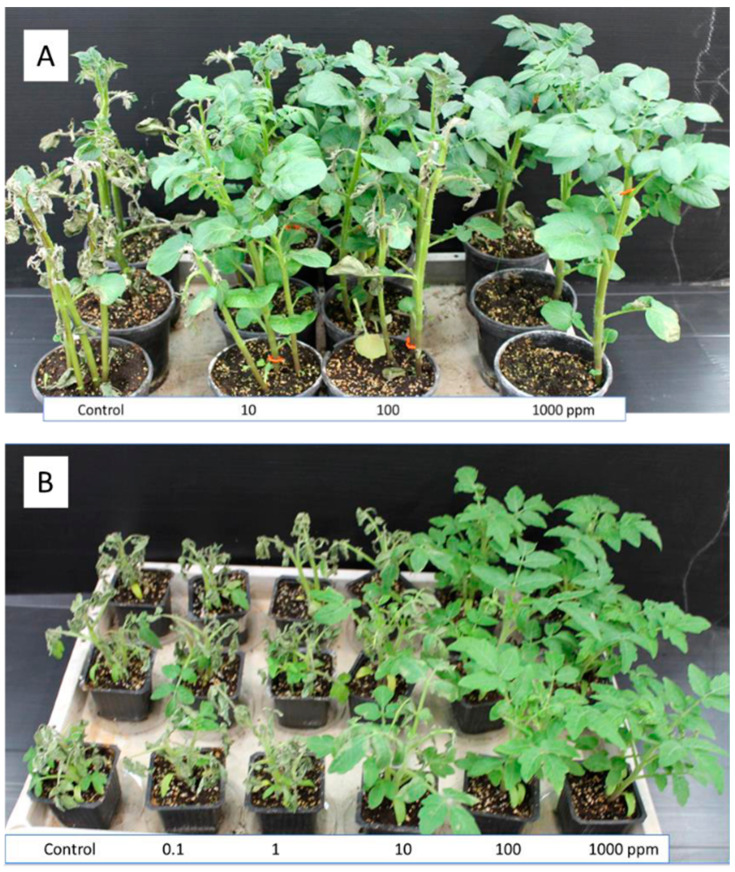
Efficacy of Zorvek Endavia (ZE, oxathiapiprolin + benthiavalicarb) in controlling late blight in potato (**A**) and tomato (**B**). Plants were sprayed with various doses (ppm product) of ZE on their upper leaf surface and inoculated 3 h later with *Phytophthora infestans*. Photos were taken at 7 dpi.

**Figure 2 jof-09-01046-f002:**
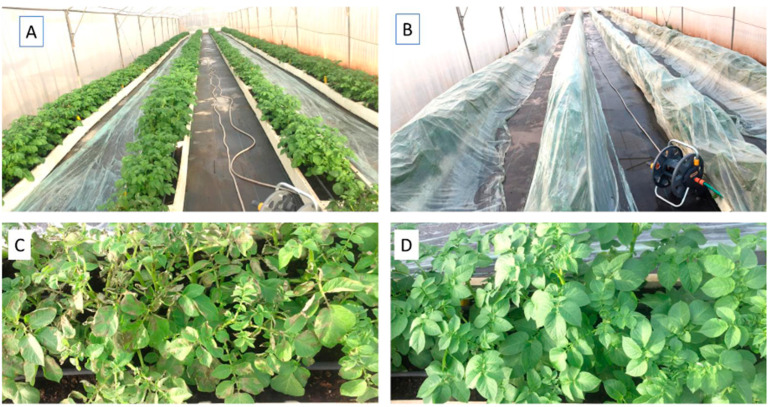
Tunnel Experiments. Appearance of the potato plants: (**A**) after fungicide application. (**B**) after inoculation. Plants were covered with plastic sheets for 14 h after inoculation. (**C**) symptoms of late blight in control plots at 6 days post-inoculation. (**D**) No late blight symptoms in fungicide-treated plots at 6 days post-inoculation.

**Figure 3 jof-09-01046-f003:**
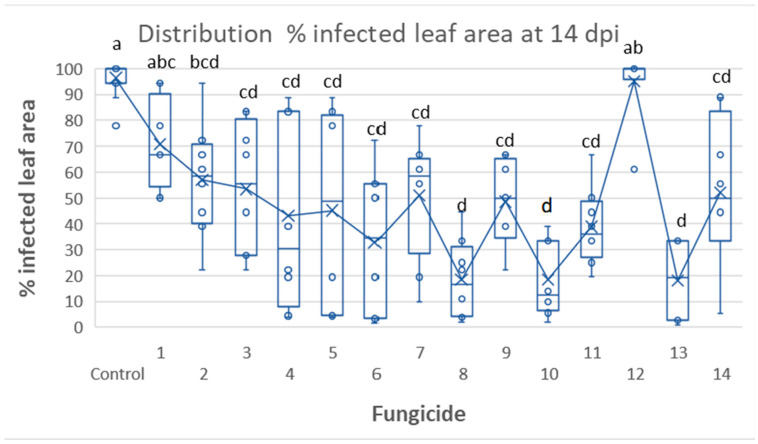
Tunnel Experiment 1: Efficacy of 14 fungicidal products against late blight in potatoes. Distribution of % infected leaf area at 14 dpi in plots treated with 14 fungicides. Different letters above the columns indicate significant differences between means (Tukey HDS test, α = 0.05).

**Figure 4 jof-09-01046-f004:**
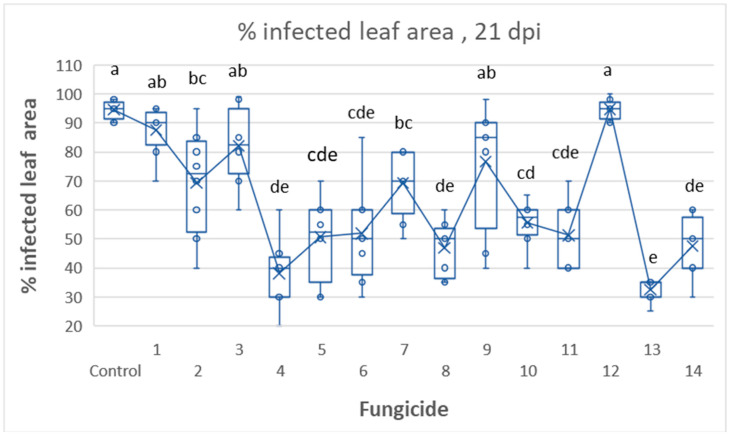
Tunnel Experiment 1: Control of late blight in potatoes by 14 fungicidal products. Distribution of disease intensity values among each fungicidal product at 21 dpi. Different letters on columns indicate significant differences between means (Tukey HDS test, α = 0.05).

**Figure 5 jof-09-01046-f005:**
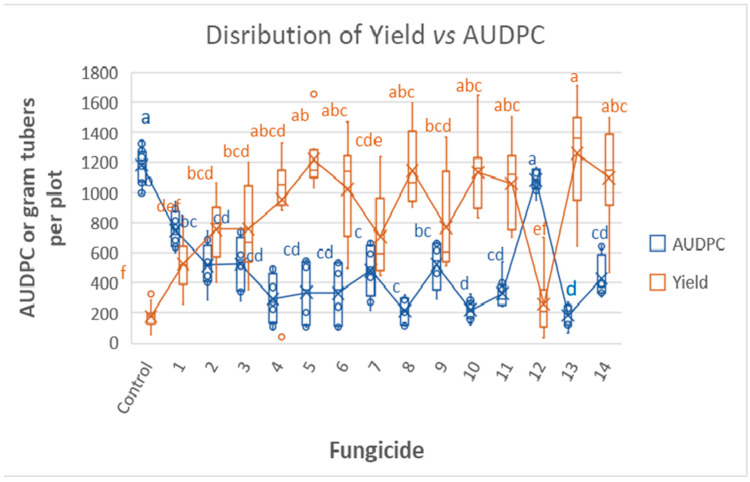
Tunnel Experiment 1: Efficacy distribution of 14 fungicides in controlling late blight in potatoes. Box chart showing the distribution of yield at 55 days after sowing vs. AUDPC of 21 days. Different letters on bars indicate significant differences between means (Tukey HDS test, α = 0.05).

**Figure 6 jof-09-01046-f006:**
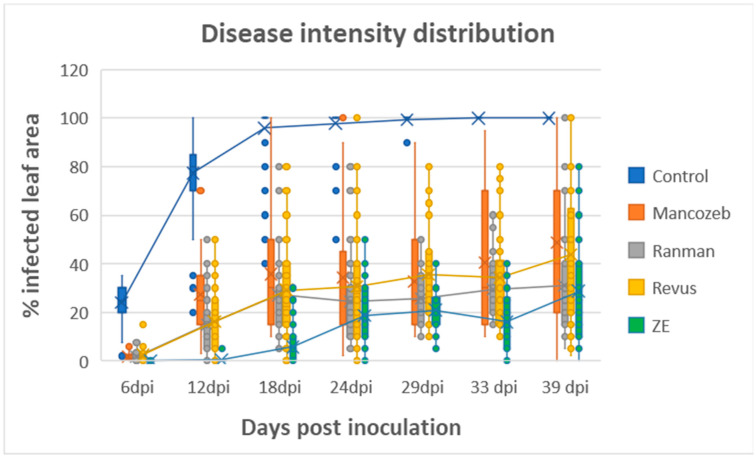
Tunnel Experiment 2. Efficacy of 4 fungicides in controlling late blight in potatoes Fungicides were sprayed throughout the season, as shown in Table 2. Mancozeb and Ranman were each applied six times at 7-day intervals; Revus was applied four times at 9-day intervals; and ZE was applied twice at 21-day intervals. The graph represents the disease severity distribution among control and the four fungicides during the 39 days of the epidemic. The Tukey HDS test (α = 0.05) at 39 dpi assigned the letters A, B, B, C, and C to plots treated with none, MZ, REV, Ran, and ZE, respectively.

**Figure 7 jof-09-01046-f007:**
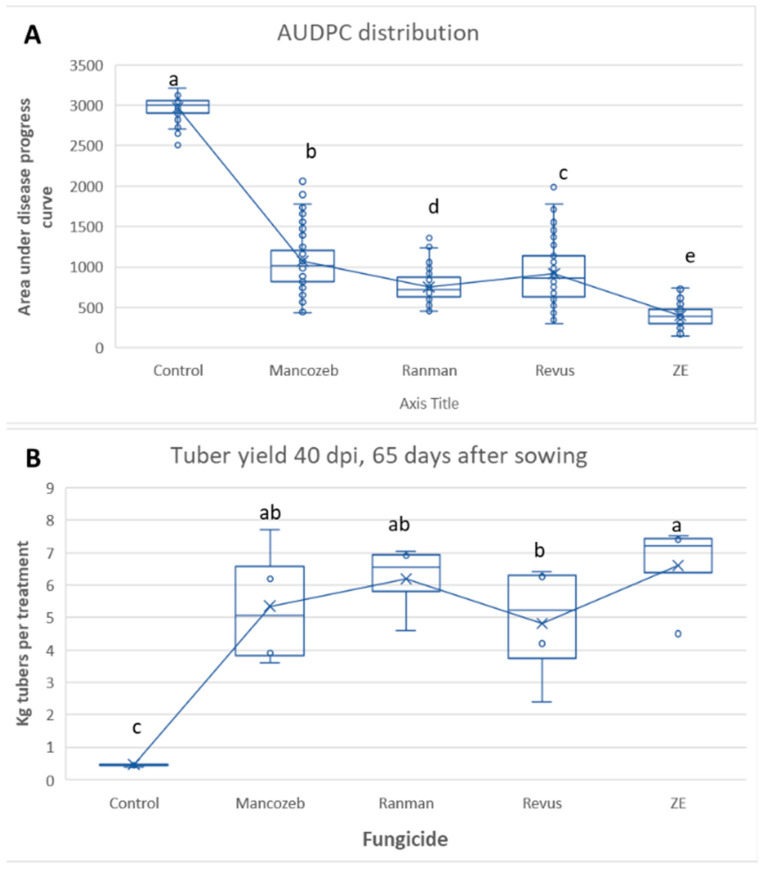
Tunnel Experiment 2. Box charts show the efficacy of four fungicides in controlling late blight in potatoes. Fungicides were sprayed throughout the season, as shown in Table 2. Mancozeb and Ranman were each applied six times at 7-day intervals; Revus was applied four times at 9-day intervals; and ZE was applied twice at 21-day intervals. (**A**) Distribution of AUDPC (area under progress curve). (**B**) Tuber yield per treatment at 65 after sowing. Different letters on the bars indicate a significant difference between treatments (Tukey HDS test at α = 0.05).

**Table 1 jof-09-01046-t001:** A list of the fourteen fungicidal preparations used in Experiment 1 of this study and their doses applied to control *P. infestans* in potato crops under tunnel conditions.

Code	Product	Active Ingredients gr/Kg	Chemical Group *	FRAC Group **	gram/h	Dose, %
**1**	AS 121	Valifenalate 60 + CuOH 150 + CuOCl 150	CAA + M	H5	200	0.4
**2**	Infinito	Fluopicolide 62.5 + Propamocarb 625	Benzamide + Carbamate	B5 + F4	150	0.3
**3**	Cabrio	Dimethomorph 72 + Pyraclostrobin 40	CAA + QoI	H5 + C3	250	0.5
**4**	Carial Plus	Cymoxanil 180 + Mandipropamid 250	CO + CAA	UN + H5	60	0.12
**5**	CYO 719	Fluazinam 300 + Cymoxanil 200	Phenyl-pyridineamine + CO	UN + C5	100	0.2
**6**	Banjo	Fluazinam 500	Phenyl-pyridineamine	C5	100	0.2
**7**	Banjo Forte	Fluazinam 200 + Dimethomorph 200	Phenyl-pyridineamine + CAA	C5 + H5	50	0.1
**8**	Revus	Mandipropamid 250	CAA	H5	60	0.12
**9**	Electis	Cymoxanil 390 + Zoxamid 330	CO + Benzamide	UN + B3	45	0.1
**10**	Ranman	Cyazofamid 500	QiI	C4	50	0.1
**11**	Mancozeb	Mancozeb 750	M	M	300	0.6
**12**	Polyram	Zineb700	M	M	300	0.6
**13**	Zorvec Endavia	Oxathiapiprolin 30 + Bethiavalicarb 70	OSBPI + CAA	C9 + H5	50	0.1
**14**	AGF 273	Fluopicolide 200 + Cymoxanil 240	Benzamide + CO	B5 + UN	60	0.12

* CAA = carboxylic acid amide; CO = cyanoacetamide-oxime; M = multisite inhibitor; OSBPI = oxysterol-binding protein homolog inhibitor; QiI = quinone inside inhibitor; QoI = quinone outside inhibitor; ** Fungicide Resistance Action Committee (https://www.frac.info/, accessed on 20 October 2023).

**Table 2 jof-09-01046-t002:** Fungicides and spray schedules were used in Experiment 2 to control late blight in tunnel-grown potatoes.

Fungicide	Kg/h	Sprays	Interval	Dates of Spray					
Mancozeb	3	6	7 days	8.12.22	15.12.22	22.12.22	29.12.22	5.1.23	12.1.23
Ranman	0.5	6	7 days	8.12.22	15.12.22	22.12.22	29.12.22	5.1.23	12.1.23
Revus	0.6	4	9 days	8.12.22	16.12.22		25.12.22	2.1.23	
ZE	0.5	2	21 days	8.12.22			29.12.22		

## Data Availability

The data that supports the findings of this study is available from the corresponding author upon reasonable request.

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
