# Peer review of "Replacing Mancozeb with Alternative Fungicides for the Control of Late Blight in Potato"

_jof, 2023, doi:10.3390/jof9111046_

Round 1

Reviewer 1 Report

This paper relates experimental studies to replace Mancozeb, a fungicide banned by the EU since January 2022, by other fungicides to fight potato and tomato late blight caused by Phytophtora infestans. On the twenty-seven fungicides initially selected and tested in growth chamber (data not reported in the paper), 14 fungicides were tested in this study. The first experiment has shown that only three fungicides were efficient and these three fungicides were further tested in a second series of experiment to determine the best strategy. The conclusion is that the three fungicides (Zorvec Endavia, Ranman and Revus) can be used alternatively to control the disease and to avoid the appearance of fungal resistance.

I have two main comments and a few minor ones.

The authors say that the two experiments are field trials. It was not conducted in open field. Potatoes were seeded in containers filled with a mix of compost and peat and covered by white plastic. These conditions allowed the authors to control the inoculation and spread of the disease and this is a good approach to evaluate the efficacy of the fungicides, however these experiments should not be named field trials, but rather tunnel trials. The paper title and corresponding paragraphs should be modified accordingly.

The statistical analysis performed in this study is a one-way analysis of variance and student t-test. Student t-test is used to compare only two sets of data. It is the same for the Fisher test used to compare 2 variances. So the authors can compare each fungicide to Mancozeb, but they cannot compare all the fungicides together with these tests. They should have used a post-hoc test like Student-Newman-Keuls (SNK). I agree that the letters on the graph are coherent with the data, however from a scientific point of view the statistical tests should be conducted with the appropriate tests.

Minor comments:

L.7: the control “of” plant diseases

L.8: “caused” instead of “incited”

L.31: Ref [1] could be replaced by a more appropriate reference such as: Fry, W. Phytophthora Infestans: The Plant (and R Gene) Destroyer. Mol. Plant Pathol. 2008, 9, 385–402, doi:10.1111/j.1364-3703.2007.00465.x.

L.40-41: “ETU, a group B2 carcinogen, was found to be the drinking water stressor resulting from the use of EBDCs.” What do you mean by water stressor? Isn’t it rather water pollutant?

L.43-46: The sentence is not clear. I propose: Mancozeb bear the potential to “be the” cause of several health problems or “Mancozeb bear the potential to can cause of several health problems…” “demonstrating with due to (?) an increase in ETU dosages”

L.82-83 The last sentence is a result and should not be present in the introduction section.

$3.1: A table with all the results should be added.

L.162: “They showed up…” Do you mean “They were observed”

Figure 2A: some colors are very similar and thus difficult to differentiate. I suggest for example to replace the dots by squares in samples 10 to 14.

L.261: “They compose” should be replaced by “They are composed”.

L.264-267: could you give the formula used to calculate the efficacy in the method section?

References:

Please add internet link (if available) for references 1, 2, 3 and 12

Author Response

see attached doc

Reviewer 2 Report

Main and Cohen provide a short paper on fungicidal treatments to control Phytophthora infestans in potato. The purpose of the study was to find an efficacious fungicidal alternative to the commonly used fungicide mancozeb. The study is generally well written, and the experimental system is fit for purpose. However, I think the authors need to make major changes to the manuscript before it is acceptable for publication in JoF. I have added my main comments below, and also attach an annotated PDF with my specific comments. I hope the authors find my comments useful.

Introduction

Your introduction needs more information on previous studies assessing fungicide efficacy against P. infestans. Im sure there is plenty.

You deal with a lot of information about the risks of MZ, and how the EU is banning it. I think this part about human and environmental risks can be significantly reduced to a sentence or maybe two. 

Throughout the paper you need to make sure that if you abbreviate a term then you should only use the abbreviation in the rest of the manuscript. Many times you spell out the full terms for (Mode of action, mancozeb) when you should just write MOA or MZ. You should also abbreviate Phytophthora infestans to P. infestans anytime after your first mention of it.

Methods

You need to give specific names to your three experiments, and stick with these names. It is confusing when you refer to experiment 1 as the first field experiment. I would say that the chamber experiment was actually your first experiment. You should call them “growth chamber experiment, field experiment 1, field experiment 2. Use these terms throughout the manuscript.

Its not clear how disease was assessed in the field experiment1 and 2.

Your data section is missing some info on yield measurements and on correlation analyses

Results

You have much too many graphs. Many of the graphs are just showing the same data. I have suggested you delete many graphs, and keep the bar and whisker plots, but include the statistically significant letters on these bar and whisker plots.

Wy do you deal with results from two different time points in the experiment? Surely its just the amount of disease at the end of the experiment that matters, and not disease progression over time.

Discussion

Your discussion does not put your results into context. It often just reiterates your results. You need to provide context by referring to other studies, and showing how your results are similar/different.

You also need to reference other studies that tested fungicides for potatoes against P. infestans. You study is certainly not the first to do this.

Generally very good, a few small typos

Round 2

Reviewer 1 Report

The manuscript was significantly improved; however, discussion should still be improved. It is mainly a repetition of the results without comparison (or a few) with other studies.

Minor comments:

L.18: change to: 39 days post inoculation (dpi)

L.37: million dollars?

L.113 and 147: remove “field” before “tunnel”

Table 1: add the information about FRAC as requested by reviewer 2: Fungicide Resistance Action Committee (https://www.frac.info/).

L.168-169: Include potato yield into paragraph 2.4. As yield results are presented, this sentence should be maintained

L.193-194: replace “days post infection” by “dpi”.

L.196: remove “days post infection” and parenthesis around “dpi”

Figure 2B: the photos are not really useful as plants are covered with plastics… This step is not described in the methods. Please add that step in the methods. Where the plastics removed totally after 14h?

Figure 3: there is no letter for treatment 10.

L.216 and 253-257: The authors present data on tuber yield in figure 5, so it should be cited in the methods.

Figure 5: Check letters. Letters “c” in blue probably correspond to products 6 and 7 and there should be a “d” above product 8. There is no letter for product 13 (d?).

L.249: remove “area under disease progress curves” and the parenthesis around AUDPC.

Figure 6: Disease is not written properly in the graph title. Mancozeb has a k (Mankozeb) in the legend.

Figure 7: A and B letters are not present above the graphs. However why did the authors separate the two graphs instead of representing it like in Figure 5? Significant letters were not reported. Add “days” after “65” L.271.

L.276: “.” Lacks between MZ and Farmers.

L.274-284: Already said in the introduction, not useful for the discussion.

L.304-306: the same thing is said just above L.297-299.

L.324: It would be nice to be more precise: what is the ratio between the increased cost and the reduction of the number of sprays?

L.353: add internet link: https://cpb-us-w2.wpmucdn.com/u.osu.edu/dist/b/28945/files/2020/02/frac-code-list-2020-final.pdf

L.355: add internet link: http://www2.mda.state.mn.us/webapp/lis/chemsold_default.jsp

L.359: Write mancozeb in full, not MZ in the reference section and add internet link: https://archive.epa.gov/pesticides/reregistration/web/pdf/mancozeb_red.pdf

L.377: add internet link: https://www.fas.usda.gov/data/european-union-mancozeb-non-renewal-and-mrl-review

L.378: add internet link: https://eur-lex.europa.eu/legal-content/EN/TXT/?toc=OJ%3AL%3A2021%3A160%3ATOC&uri=uriserv%3AOJ.L_.2021.160.01.0089.01.ENG

Author Response

see attached doc

Reviewer 2 Report

please see below some more comments. thanks for taking my previous points on board.

Page 8 figure 3- misspelling in title of figure (distribution)

Page 8 figure 3- no letter over bar for fungicide 10

Page 9 figure 5 – misspelling of distribution in figure title

Page 9 figure 5 difficult to read letters over treatment bars 2, 3. No letter over bar for treatment 13

Page 10 figure 6 misspelling title Disease

Page 10 not clear that figure 7 is a 2 panel (A, B) figure and that the figure legend on page 11 refers to the AUDPC figure above

some poor spelling in the figures. please double check all as they wont be picked up by spellcheck

Author Response

see attached doc
